# Tapered, Twisted Bundled-Tube Locomotive Devices for Stepped Pipe Inspection

**DOI:** 10.3390/s22134997

**Published:** 2022-07-02

**Authors:** Daisuke Shiomi, Toshio Takayama

**Affiliations:** Department of Mechanical Engineering, Tokyo Institute of Technology, 2-12-1, Ookayama, Tokyo 152-8552, Japan; takayama.t.aa@m.titech.ac.jp

**Keywords:** soft robotics, in-pipe, pneumatic, snake-like, helical

## Abstract

The twisted bundled-tube locomotive device is an elongated soft robot that moves inside a pipe in a helical bending motion. This motion mimics the behavior of microorganisms called spirochetes. This device is inexpensive and easy to miniaturize because of its simple structure, which consists of three inflatable tubes twisted together. It can move in pipes of various diameters without a change in design. Therefore, it has a high capacity for water pipe inspection. However, it has not yet been shown to pass through step parts wherein the diameter of the pipes decreases. In this study, we developed a device that was deformed into a tapered shape by changing the pitch of the spirals at each location. The prototype device was able to move from a pipe with an inside diameter of 52.9 mm to a pipe with an inside diameter of 21.6 mm for horizontally fixed pipes, and from a pipe with an inside diameter of 41.6 mm to a pipe with an inside diameter of 21.6 mm for vertically fixed pipes.

## 1. Introduction

Although snakes have a simple structure with a cord-like body, they can move on uneven ground. They also sometimes use their bodies dexterously to perform various actions such as climbing trees. Therefore, despite their simple structure, many robots that mimic snake behavior have been studied due to the fact that various movements are possible [1,2,3,4,5,6,7]. Robots that mimic the motion of snakes do not require an infinite rotating axis such as a wheel because they can generate thrust with only a bending motion. This makes it easier for them to seal and protect the interior of the robots from water and gas. In addition, their elongated structure allows them to move within narrow spaces. These advantages allow them to be used in a variety of environments, including dirty areas, inside pipes, and underwater environments. For example, several studies have been conducted on swimming snake-like robots that mimic the bending motion of sea snakes [8,9,10]. Several studies have also been conducted on snake-like robots for movement inside pipes [1]. Most snake-like robots utilize a planar bending motion. Snakes produce this type of movement owing to the effects of gravity. However, some microorganisms drifting in water perform three-dimensional bending motions. One of these is a spiral-shaped microorganism called a spirochete. It generates a propulsive force by rotating around its central axis, and it moves in the direction of the central axis. The three-dimensional bending motion keeps the trunk in a spiral shape and generates a propulsive force through an infinite rotational motion similar to that of a screw such that all parts of the trunk can contribute to the generation of the propulsive force in water. We thus developed an underwater swimming robot that mimics this motion [11]. This three-dimensional bending motion has also been applied to robots that move by wrapping themselves around pipes [12,13] and robots that move inside pipes. Snake-like robots can move with high efficiency by performing three-dimensional bending motions if they can move with all joints in equal contact to an environment, such as inside or outside the pipes.

In-pipe robots that are usually proposed can be categorized into wheel, screw, worm-like, and snake-like robots, depending on their movement method. Their characteristics are summarized in Figure 1.

Wheel-type robots have the advantages of high traveling accuracy and speed [14]. However, it is difficult to accommodate a wide range of pipe diameters without changing the design because the wheels must adhere to various directions for stable travel in the pipe [15]. In addition, its complex structure makes miniaturization difficult. Robots that move by screw rotation can be driven using a single actuator. Therefore, they are more energy-efficient than wheel-type robots. However, they are also not compact, and for the same reason as that for wheel-type robots, accommodating a wide range of pipe diameters in a single body is difficult [16,17]. Robots that use a peristaltic motion, similar to worms, have high sealability. However, if the device is composed of mechanical links and is actuated by electric motors, its size will increase and it requires complex controls [18]. It is difficult to miniaturize any device that uses a motor to move inside a pipe, and the complex structures of these devices make them expensive. Therefore, several soft robots have been studied as inexpensive and easily miniaturized robots [19,20]. Worm-like robots can be made inexpensively with elastic materials. By inflating their body, they can adapt to different pipe diameters. We thus propose a snake-like soft robot. Snake-like robots composed of mechanical links require many motors, which make it expensive and unreliable. Our proposed robot can overcome these problems. On the other hand, the elastic material makes it less durable. Our proposed robot can be mass-produced because of its construction.

To overcome these problems, we propose a twisted bundled-tube locomotive device. As shown in Figure 2a, this device consists of three twisted and inflatable silicone tubes. Pressurizing a single tube deforms the entire device into a helical shape, with the pressurized tube moving outward, as shown in Figure 2b. By repeatedly pressurizing and depressurizing the three tubes, the device twists and moves inside the pipes by means of friction with the pipe while maintaining its deformed helical shape.

Because a non-stretchable axis exists in the center of the device, it can be equipped with electric cables. Thus, a camera can be attached to function as an endoscope. Moreover, this device has the same cross-section as the axis. Therefore, mass production by extrusion is possible [21]. It has been demonstrated that this device can move inside pipes of various diameters without changing its structure [22]. A problem that persists for this device is that it can’t pass through step parts where the diameter of pipes decreases [23]. Robots targeted at passing through the step parts have been studied, but their complex structures make miniaturization difficult [24]. Therefore, the purpose of this study is to produce a twisted bundled-tube locomotive device that can pass through step parts without the need for complex control.

## 2. Method

### 2.1. Analysis

Consider a device model with one helix cycle that focuses on the body axis and one of the three silicone tubes that helically wrap around the body axis, as shown in Figure 3. Figure 3a shows the model before deformation, and Figure 3b shows the deformed model after air pressure is applied to the silicone tube to expand it, such that the axial expansion ratio becomes α. For simplicity, the diameter of the body axis is assumed to be negligibly small, and the silicone tube is assumed to expand only in the axial direction. The pitch of the silicone tube wrapped around the body axis before deformation is $P_{0}$ (hereafter called the initial pitch), the pitch of the helix after deformation is $P_{1}$, and the distance between the central axis of the device and center of the silicone tube after deformation is R. The distance between the body axis and the center of the silicone tube is *r*, and the angle of the spiral of the silicone tube around the body axis is γ; because the body axis and silicone tube are fixed, *r* and γ are both constant.

Based on this relationship, the device model is then developed into a plane in Figure 4. Figure 4a,b show the device before and after deformation, respectively. The lengths of the silicone tubes before and after deformation are $L_{1}$ and $L_{3}$, and that of the body axis after deformation is $L_{2}$.

The deformation of the device can be analyzed based on geometric relationships. Initially, *d* is the diameter of the silicone tube, and *D* is the diameter of the helix in the center of the silicone tube located outside after deformation. From Figure 3a, d=2r, and from Figure 3b, D=2R; because *r* and γ are constant, Equation (Equation 1) is valid. Hereafter, $L_{1}^{\prime }$ denotes the product of $L_{1}$ and α; when γ is expressed before and after deformation, Equation (Equation 2) is valid.
(1)$$P_{0}:L_{1}\times \alpha =L_{2}:L_{3}$$
(2)$$\gamma =\arctan\left(\frac{\pi d}{P_{0}}\right)=\arctan\left(\frac{\pi D}{P_{1}}\right)-\arctan\left(\frac{\pi (D-d)}{P_{1}}\right)$$
$L_{1}$, $L_{2}$, and $L_{3}$ from Figure 4 are expressed as in Equations (Equation 3)–(Equation 5). Substituting $L_{2}$, and $L_{3}$ into Equation (Equation 1) and solving for $P_{1}$ results in Equation (Equation 6). Equation (Equation 2) can be solved for $P_{1}$ to obtain Equation (Equation 7).
(3)$$L_{1}=\sqrt{P_{0}^{2}+{\left(\pi d\right)}^{2}}$$
(4)$$L_{2}=\sqrt{P_{1}^{2}+\pi^{2}{\left(D-d\right)}^{2}}$$
(5)$$L_{3}=\sqrt{P_{1}^{2}+{\left(\pi D\right)}^{2}}$$
(6)$$P_{1}=\pi \sqrt{-D^{2}+\left(\frac{2dDL_{1}^{\prime 2}}{L_{1}^{\prime 2}-P_{0}^{2}}\right)-\left(\frac{d^{2}L_{1}^{\prime 2}}{L_{1}^{\prime 2}-P_{0}^{2}}\right)}$$
(7)$$P_{1}^{4}+P_{1}^{2}\left(2\pi^{2}D^{2}-2\pi^{2}Dd-P_{0}^{2}\right)+\pi^{4}D^{2}{\left(D-d\right)}^{2}=0$$

Finally, substituting Equation (Equation 6) into Equation (Equation 7) and solving for *D* yields Equations (Equation 8) and (Equation 9).
(8)$$D=\frac{-C_{1}+\sqrt{C_{1}^{2}-4C_{2}C_{0}}}{2C_{2}}$$
(9)$$\left\{\begin{array}{c} C_{2}=\pi^{2}d^{2}{\left(\frac{L_{1}^{\prime 2}+P_{0}^{2}}{L_{1}^{\prime 2}-P_{0}^{2}}\right)}^{2}+P_{0}^{2}\\ C_{1}=-\frac{2\pi^{2}d^{3}L_{1}^{\prime 2}\left(L_{1}^{\prime 2}+P_{0}^{2}\right)+2dP_{0}^{2}L_{1}^{\prime 2}\left(L_{1}^{\prime 2}-P_{0}^{2}\right)}{{\left(L_{1}^{\prime 2}-P_{0}^{2}\right)}^{2}}\\ C_{0}=\frac{\pi^{2}d^{4}L_{1}^{\prime 4}+d^{2}P_{0}^{2}L_{1}^{\prime 2}\left(L_{1}^{\prime 2}-P_{0}^{2}\right)}{{\left(L_{1}^{\prime 2}-P_{0}^{2}\right)}^{2}} \end{array}\right.$$

The radial expansion of the silicone tube is considered here. The radial expansion rate is β, and A is the outer diameter of the device after deformation. A can be expressed as shown in Equation (Equation 10).
(10)A=D+βd

The relationship between α and β for the silicone tubes used in this study can be approximated by using Equation (Equation 11) [25].
(11)(β−1)=0.60(α−1)

From these equations, the outer diameter A of the device after deformation can be obtained from the initial pitch $P_{0}$ and the axial expansion rate of the silicone tube α. Similar analyses can be found in references [26,27]. From our experience, the silicone tube used in this study is easily punctured when the applied pressure and alpha exceed 0.14 MPa and 1.11, respectively. Therefore, we used α = 1.11 to design the device.

### 2.2. Proposal of the Tapered Shape

Figure 5 shows the relationship between the initial pitch and the outer diameter after deformation, when α = 1.11 and using Equations (Equation 8)–(Equation 11). Figure 5 shows that a larger initial pitch results in a larger outer diameter after deformation, whereas a smaller initial pitch results in a smaller outer diameter after deformation. Figure 6a illustrates the deformation at the initial pitch and the expansion rate shown in Figure 5a, and Figure 6b illustrates the deformation shown in Figure 5b. If the outer diameter after deformation is small, as shown in (a), it cannot generate a pushing force against the inner wall of the larger pipe to generate a gripping force, and thus it cannot move. If the outer diameter after deformation is large, as shown in (b), it can generate a pushing force against the inner wall of the larger pipe and can thus move. However, the front tip is always pressed against the inner wall; therefore, it is caught by the step part and cannot enter the smaller pipe.

To pass through the step part, we propose a device that deforms into a tapered shape that is thicker at the root and thinner toward the tip when air pressure is applied, as shown in Figure 6c. A device with this tip shape is believed to pass through the step part because the tip is pointed and can move freely in the radial direction. A tapered tip shape is produced by bundling the root with a helix possessing a larger pitch, and then bundling it with a helix possessing a smaller pitch as it approaches the tip. We aim to move the step part by driving a twisted bundled-tube locomotive device with this tip geometry under appropriate pneumatic control.

## 3. Experiments

### 3.1. Tube Manufacturing Method

In Japan, the 20A pipe is widely used for water pipes, with the designation A being the nominal diameter of a water pipe, as defined by the Japanese Industrial Standard (JIS). The inside diameter of a 20A pipe is 21.6 mm. Considering the deformation of the device, the device was designed such that its diameter is approximately half of the inside diameter of a 20A pipe. Therefore, silicone tubes with an outer diameter of 6 mm were used to fabricate it. A commercial silicone tube is not designed to be inflated and can only be inflated at high pressure. In addition, it does not return to its original size. Therefore, soft silicone tubes have been fabricated. Molds and caps were used, as shown in Figure 7a. The molds had half pipe-shaped grooves. The diameter of these grooves was the same as that of the developed silicone tube. The caps were shaped to fit both ends of the mold. They also had a hole in the center through which a stainless-steel rod could be passed. The molding proceeded as follows: (1) Liquid silicone was poured into the grooves of the molds, as shown in Figure 7b. (2) Stainless-steel rods were prepared. These rods should be slightly longer than the mold and slightly thicker than the commercial tubes joined at both ends of the silicone tube. (3) Both ends of the stainless-steel rod were inserted approximately 1 cm into the commercial tubes. A cap was then placed at both ends. (4) A stainless-steel rod was sandwiched between the molds, as shown in Figure 7c. When liquid silicone hardens, it merges with the commercial tubes. Once hardened, as shown in Figure 7d, the silicone tube was removed from the molds and caps, and the rod was pulled out. In this way, silicone tubes with a length of 220 mm, an outer diameter of 6 mm, and an inner diameter of 1.6 mm were fabricated with X-32-2428-4 (Shin-Etsu Chemical Co., Ltd., Tokyo, Japan).

### 3.2. Experiment Using a Uniform Pitch Device

For comparison, an experiment was conducted using a device with a constant initial pitch. From the analysis results, the relationship between α and the outer diameter after deformation can be expressed by determining the initial pitch. Based on Equations (Equation 8)–(Equation 11), a graph that shows the relationship between α and the outer diameter after deformation was drawn. Several graphs were drawn by changing the initial pitch $P_{0}$, and when α=1.11 in the graph of $P_{0}=320$, the outer diameter after deformation became 52.9 mm, which is the same as the inner diameter of a 50A pipe, as shown in Figure 8. From this result, the larger pipe used in the following experiments was determined to be a 50A pipe; the 50A pipe is also a widely used pipe standard in Japan. We manufactured a device with an initial pitch of 320 mm and attempted to move from 50A to 20A.

The device was driven by applying air pressure using three solenoid valves (VP544R-6M1-A, SMC Co., Ltd., Tokyo, Japan ) driven by a USB I/O Terminal (A10-160802AY-USB, CONTEC Co., Ltd., Osaka, Japan ). The period, duty ratio, and strength of the air pressure were controlled. The period spans the time from the start of pressurizing a tube to the start of pressurizing it again after depressurizing it (indicated by T), as shown in Figure 9. The duty ratio is the ratio of the time required to pressurize each tube to the period, indicated by $D=\frac{t}{T}$, as shown in Figure 9. The tube takes time to fully inflate. Therefore, if the time when the two tubes are simultaneously pressurized is short, a timing will exist when the pressurization is insufficient; hence, twisting cannot be performed while maintaining the deformed shape at the assumed α. In addition, a longer time for each tube to be pressurized generates time for the device to stand still. This reduces the movement efficiency. First, the duty ratio was verified. The device was driven at various duty ratios with an air pressure of 0.14 MPa (gauge pressure) and a period of 1000 ms, as shown in Figure 10. When the duty ratio was low, such as 0.38, the device could not maintain its deformed shape during twisting. At a duty ratio of 0.43, the device maintained a constant deformation shape and torsion. These results indicate that the deformed shape could be maintained when the two tubes were simultaneously pressurized for 90–100 ms.

Next, the period was verified by simultaneously changing the duty ratio and the period such that the time during which the two tubes were simultaneously pressurized was within this range. If the period is more than 700ms with a duty ratio of 0.47, the period is sufficiently long to expand the silicone tube because there is a moment when the tube is sufficiently inflated and the device stops moving, as shown in Figure 11.

Finally, air pressure was confirmed. The expansion rate of the tube depends on the strength of the air pressure. The constructed device was gradually pressurized, and when the pressure reached 0.14 MPa, the device was deformed and acquired the same outer diameter as the inner diameter of the 50A pipe, as shown in Figure 12b. Consequently, our assumption was confirmed—that when the tube is pressurized at 0.14 MPa, the expansion ratio is 1.11. Based on the aforementioned verification, the experiment was conducted at an air pressure of 0.14 MPa, within a period of 700 ms, and at a duty ratio of 0.47. Two transparent pipes, one with the same inner diameter as that of the 50A pipe and the other with the same inner diameter as that of the 20A pipe, were fabricated using a polypropylene sheet and connected such that their centers coincided and were fixed horizontally. As shown in Figure 12d, when the device was driven, the tip of the device was caught in the step part and could not pass through.

### 3.3. Design of the Proposed Device

A device was designed with the proposed tip geometry to pass from a 50A to a 20A. As before, the initial pitch at the root was designed to be 320 mm, assuming that it was driven with an α of 1.11. To move efficiently within the 50A, it must stick strongly to all directions within the pipe. Therefore, the root was wound at a constant pitch with a length of 130 mm, which is approximately one round of the spiral after deformation. The device was designed such that the initial pitch gradually decreased as it approached the tip that had an initial pitch of 50 mm. The manufactured device is shown in Figure 13a. When pressurized, the device was deformed into a tapered shape, as shown in Figure 13b.

### 3.4. Experiment Using a Horizontally Fixed Stepped Pipe

Experiments were performed using the pipe used in the previous experiment. The pipe was fixed horizontally. The device fabricated in Section 3.3 was driven within a period of 700 ms and at a duty ratio of 0.47. When the device was driven by an air pressure of 0.14 MPa, it moved through the step part. The same experiments were performed four more times, and the same results were obtained all four times, as shown in Figure 14. Next, experiments were conducted at different air pressure strengths. When the device was driven at 0.10 MPa, the tip of the device was caught by the step part, as shown in Figure 15a. When the device was driven at 0.15 MPa, the base of the device, which was deformed and acquired a large diameter, was caught by the step part, as shown in Figure 15b.

### 3.5. Experiment Using a Vertically Fixed Stepped Pipe

The pipe used in the previous experiment was fixed vertically. The device used in the previous experiment was driven within a period of 700 ms and at a duty ratio of 0.47. When the device was driven by an air pressure of 0.14 MPa, the device remained stationary inside a pipe, as shown in Figure 16a; however, once it started moving, it slid inside the pipe. By increasing the air pressure, this device could be strongly stuck inside the pipe [25]. On the other hand, even when the air pressure was increased up to 0.18 MPa, the same results were obtained, as shown in Figure 16b. Consequently, we confirmed that the prototype device could not move within the 50A. Therefore, the movement in a vertically fixed tube was evaluated using a 40A pipe, which is one thickness smaller than the 50A.

A transparent pipe with the same inner diameter as that of the 40A was manufactured and connected to a pipe with the same inner diameter as that of the 20A, such that their centers coincided and were fixed vertically. When the device was driven by an air pressure of 0.14 MPa, it remained stationary inside the pipe, as shown in Figure 17, However, once it started moving, it slid inside the pipe. When the device was driven at 0.15 MPa, it moved through the step part, as shown in Figure 18.

## 4. Discussion

First, the minimum and maximum pipe diameter within which the device can move at an air pressure of 0.14 MPa is discussed. When pressure is applied to the silicone tube, the device expands both axially and radially. From the relationship between α and β in Equation (Equation 10), the body diameter of the device can be obtained as follows: $2\times 6\times \left\{0.60(1.11-1)+1\right\}=12.792$. Therefore, we can estimate that the diameter of the smallest pipe within which the device can move is 12.792 mm. The difference between the diameters with the largest and the smallest values is calculated as follows: 52.9−12.792=40.108 (mm). In the same way, we can estimate the difference between the diameters due to α. The difference becomes maximal at α=1.06. The largest pipe diameter is 60.40 mm and the smallest is 12.432 mm, and the difference between the diameters is 60.40−12.432=47.968 (mm). We thus assume that this difference can be further increased by increasing the initial pitch since the maximum is larger.

In an experiment using a horizontally fixed pipe, the device moved when driven at 0.14 MPa, but not at 0.10 MPa and 0.15 MPa. Based on these results, the effect of air pressure strength on the device is discussed. Figure 5 indicates that the device is deformed and acquires a larger diameter when pressurized at 0.10 MPa instead of 0.14 MPa. However, this was not the case in this experiment. Hence, at a low air pressure, the friction between the device and the pipe was believed to prevent deformation, and the device could not deform as analyzed. Therefore, the diameters of the root and tip did not differ greatly, and the tip of the device was caught in the step part. When the device was driven at 0.15 MPa, it lost flexibility owing to the high air pressure and could not accommodate pipes with a smaller diameter.

We also discuss the velocity of the device. From the experimental video shown in Figure 13, we measured the times at which the device began to enter the thin pipe and when the whole body finished entering in order to obtain the average progress velocity of the device. We also measured the progress velocity of the device in the 20A and 50A pipes that have no steps inside them. Next, we calculated the theoretical progress velocities of the device. The theoretical velocity is calculated from the pipe diameter, the pitch of the device, and the driving period [22]. It is difficult to estimate the deformation while the device passes the step part because its deformation is not constant. Moreover, the initial pitch of the device is not constant. Therefore, we obtained theoretical values by applying the maximum and the minimum pitches to the equations. If there exist multiple theoretical velocities, the minimum and the maximum values are shown (Figure 19). If the theoretical velocity can be estimated by the average of the maximum and the minimum values, the progress efficiency can be calculated from 0.65 (in 20A) to 0.76 (in 50A).

In an experiment using a vertically fixed pipe, the device that was meant to move within the 50A pipe failed to achieve this. Because the device could remain stationary in the pipe, it was deformed and acquired a diameter of 52.9 mm, as designed. The slippage and fall during the drive can be attributed to the length of the device. The device has a short contact length with the inside of the pipe owing to the small diameter of the tip, which results in a low gripping force on the pipe. Increasing the length of the device and the length that contributes to its sticking allows it to move through the pipe without slipping.

Finally, we verified whether it was possible for the device to move inside a vertically fixed pipe that had the same diameter as that of the 50A pipe by using a device with a uniform initial pitch of 320 mm. When the device was driven by an air pressure of 0.17 MPa and 0.18 MPa, it sometimes moved upward in the pipe. However, it slid inside the pipe, as shown in Figure 20. When the device was driven by an air pressure of 0.20 MPa, it deformed into a helix smaller than the diameter of the pipe. Therefore, it soon slid inside the pipe, as shown in Figure 21. From the experiments, it was confirmed that moving within a vertically fixed 50A pipe is difficult for the current device design. The cause of this is a lack of gripping force. The gripping force of the device is determined from the driving pressure and the contact area [22]. The contact area is determined by the outer diameter of the device after deformation. Therefore, the gripping force can be increased by increasing the initial pitch in order to increase the outer diameter after deformation.

## 5. Conclusions

The purpose of this study was to develop a twisted bundled-tube locomotive device capable of moving over a step. A new tip shape and method for gradually changing the initial pitch were proposed. In the experiments using a horizontally fixed pipe, the device moved from 50A to 20A when driven at 0.14 MPa. The same result was obtained even after the experiment was repeated. In the experiments using a vertically fixed pipe, the device moved from 40A to 20A when driven at 0.15 MPa. However, the device could not move inside the 50A even when a uniform device whose gripping force was larger than that of tapered devices was used. In the future, we will attempt to move this device through a pipe with a larger diameter difference. Therefore, we will improve the tubes that can be applied with greater pressure and make a device with a larger initial pitch in order to achieve a more considerable deformation. Moreover, we will try to make the device with an extrusion method in order to enable mass production. The proposed method required a different pitch at the front tip. It is necessary to develop methods that can make different pitch areas by extrusion. 

## Figures and Tables

**Figure 1 sensors-22-04997-f001:**
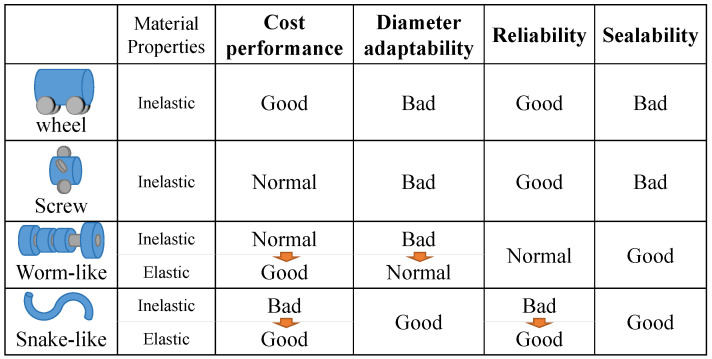
Characteristics of in-pipe robots.

**Figure 2 sensors-22-04997-f002:**
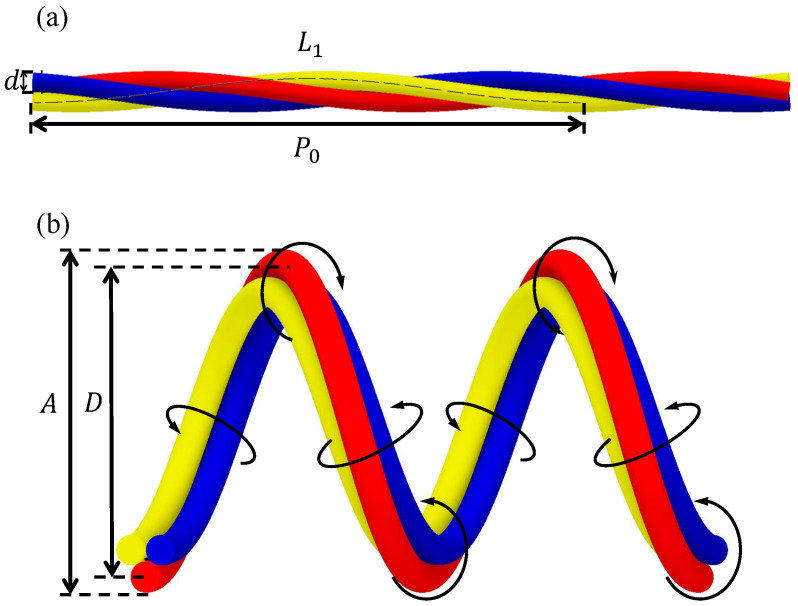
Overview of a twisted bundled-tube locomotive device. (**a**,**b**) are the initial shape and deformed shape when air pressure is applied to a single silicon tube, respectively.

**Figure 3 sensors-22-04997-f003:**
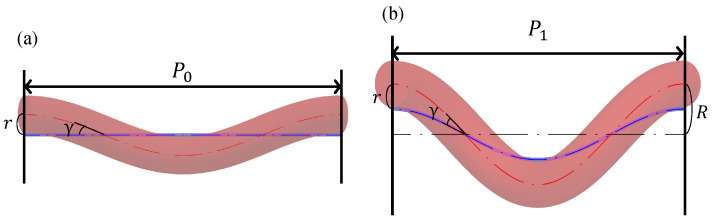
Device model with one helix cycle that focuses on the body axis and one of the three silicone tubes that helically wrap around the body. (**a**,**b**) show the device before deformation and after deformation, respectively.

**Figure 4 sensors-22-04997-f004:**
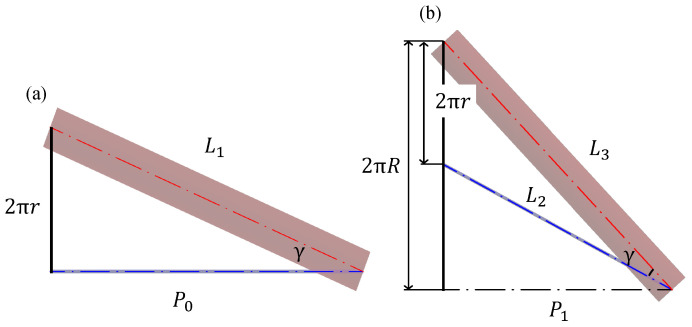
Expanded model where (**a**,**b**) show the device before deformation and after deformation, respectively.

**Figure 5 sensors-22-04997-f005:**
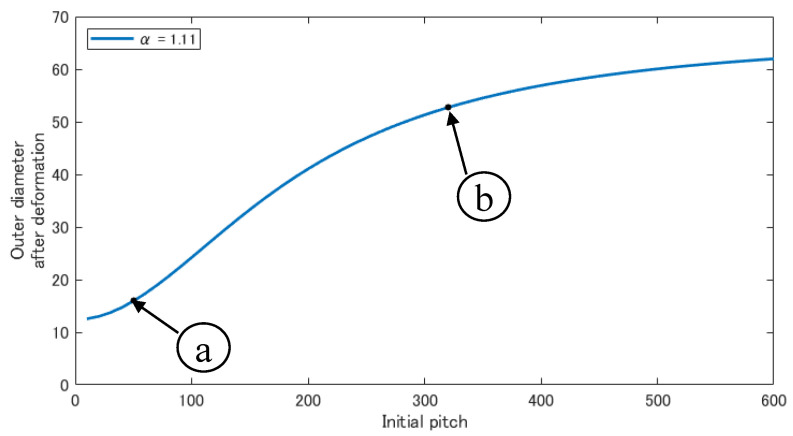
Relationship between the initial pitch and the outer diameter after deformation when α=1.11.

**Figure 6 sensors-22-04997-f006:**
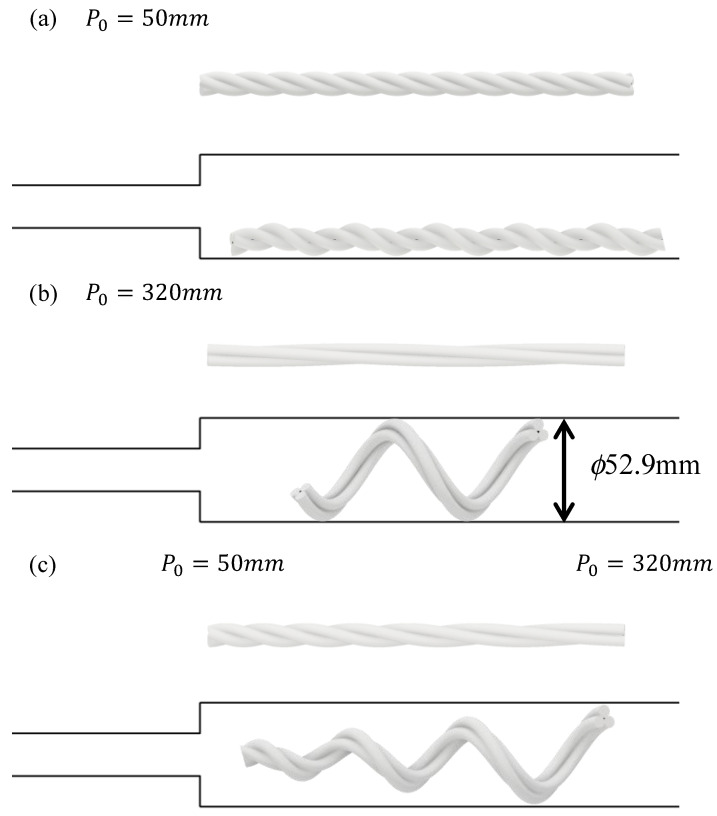
Initial shape and deformed shape when α=1.11, where (**a**–**c**) show devices whose initial pitches are 50 mm, 320 mm, and biased, respectively.

**Figure 7 sensors-22-04997-f007:**
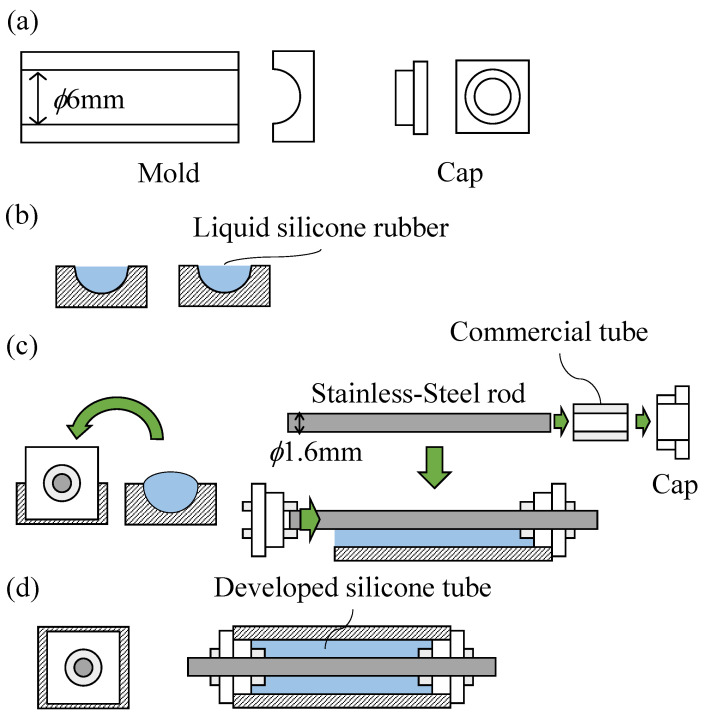
Molding procedure, where (**a**–**d**) show overview of molds and caps, liquid silicone which poured into the grooves of the molds, A stainless-steel rod which sandwiched between the molds, and the developed silicone tube.

**Figure 8 sensors-22-04997-f008:**
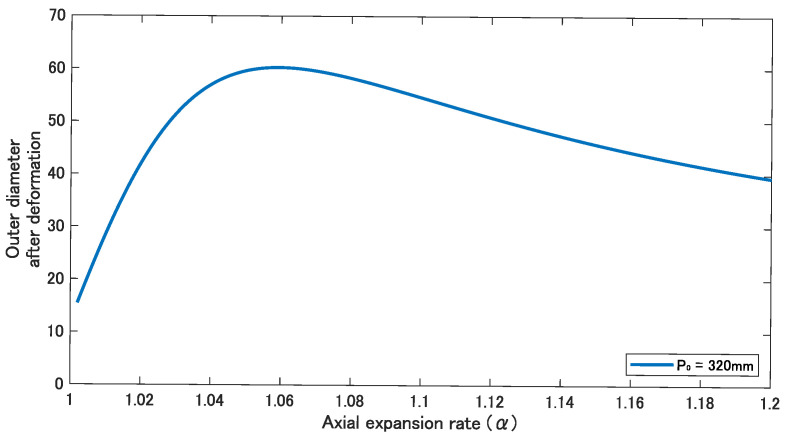
Relationship between the axial expansion rate and the outer diameter after deformation.

**Figure 9 sensors-22-04997-f009:**
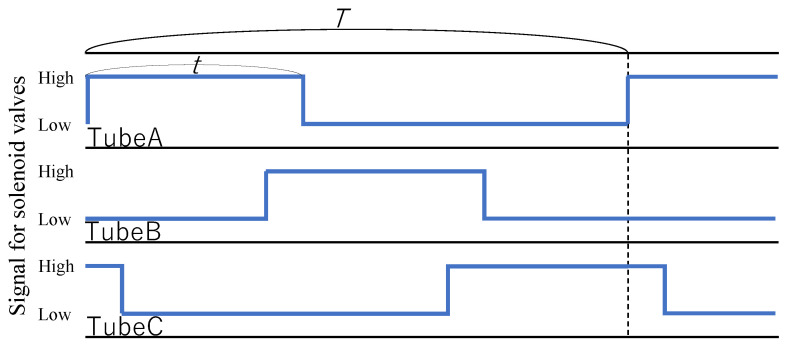
Pneumatic control.

**Figure 10 sensors-22-04997-f010:**
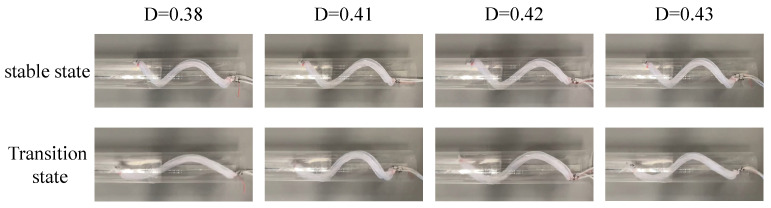
Experiments in which the device was driven within a period of 1000 ms and at an air pressure of 0.14 MPa with different duty ratios.

**Figure 11 sensors-22-04997-f011:**
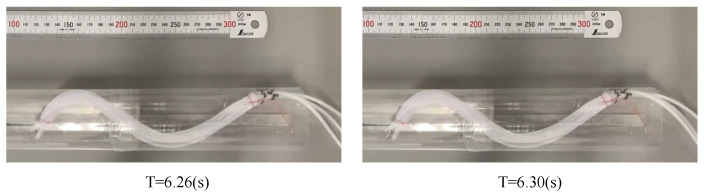
Experiments in which the device was driven with a period of 700 ms, a duty ratio of 0.47, and an air pressure of 0.14 MPa.

**Figure 12 sensors-22-04997-f012:**
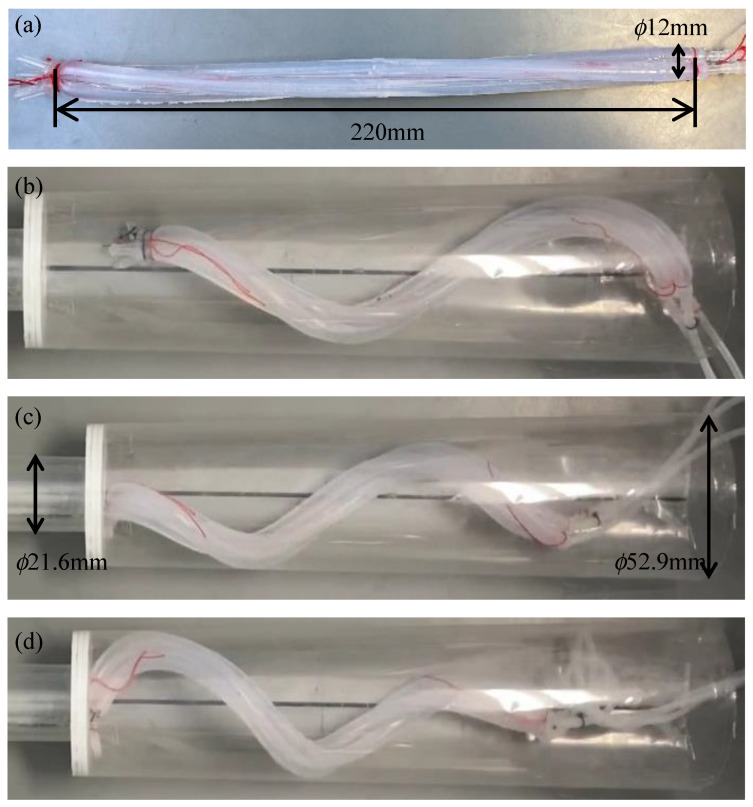
Experiment using a uniform pitch device. (**a**–**d**) show the initial shape, deformed shape, the moment when the front tip contacts the step, and the situation when the tip is caught by the step part, respectively.

**Figure 13 sensors-22-04997-f013:**
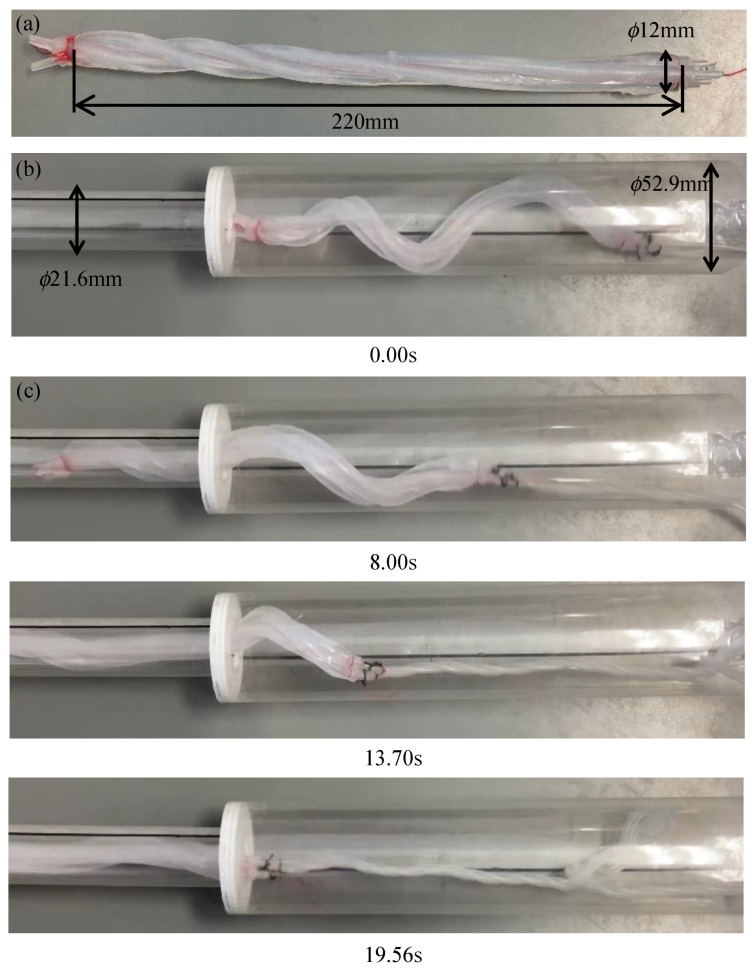
Experiment using a biased pitch device driven by an air pressure of 0.14 MPa in a horizontally fixed pipe with the same diameter as that of the 50A and the 20A pipes. (**a**–**c**) show the initial shape, the deformed shape, and the process of being inserted into a 20A pipe, respectively.

**Figure 14 sensors-22-04997-f014:**
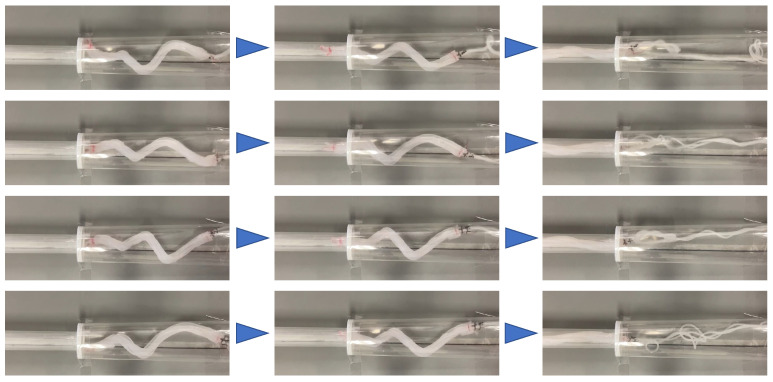
Results of four experiments conducted under the same driving conditions as in Figure 13.

**Figure 15 sensors-22-04997-f015:**
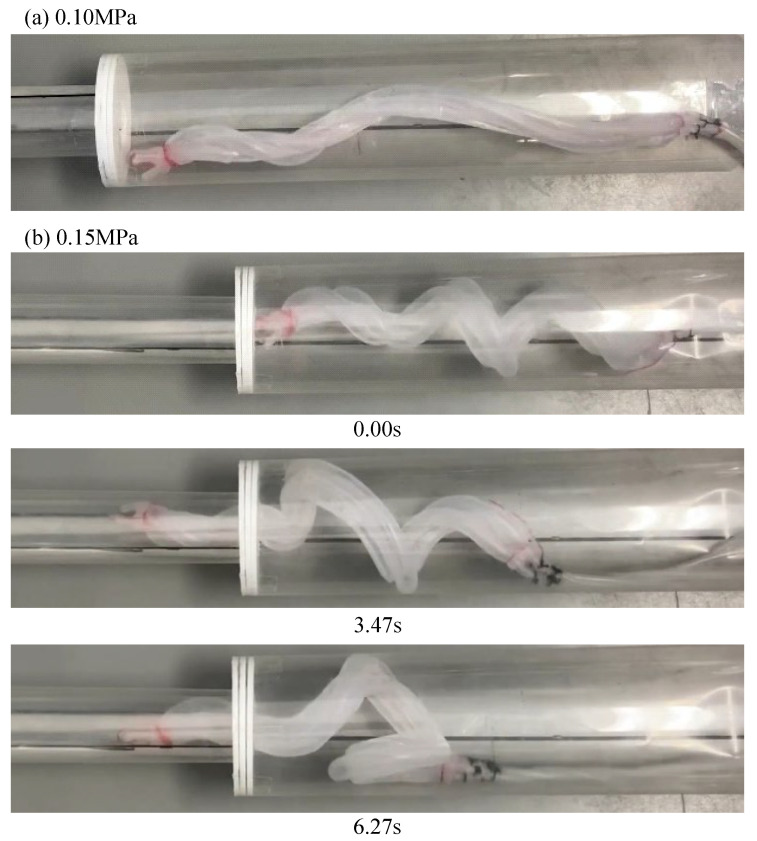
Experiment using a biased pitch device in a horizontally fixed pipe with the same diameter as that of the 50A and 20A pipes. (**a**,**b**) depict the device driven by an air pressure of 0.10 MPa and 0.15 MPa, respectively.

**Figure 16 sensors-22-04997-f016:**
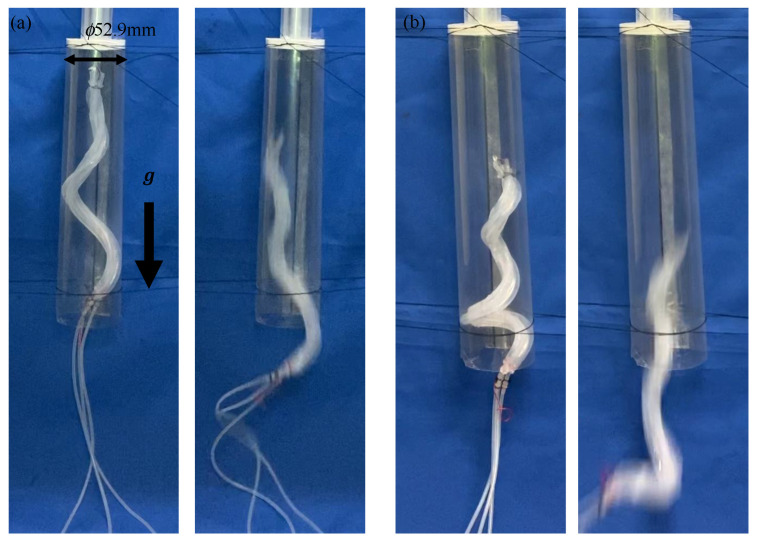
Experiment using a biased pitch device in a vertically fixed pipe with the same diameter as that of the 50A. (**a**,**b**) depict the device driven by an air pressure of 0.14 MPa and 0.18 MPa, respectively.

**Figure 17 sensors-22-04997-f017:**
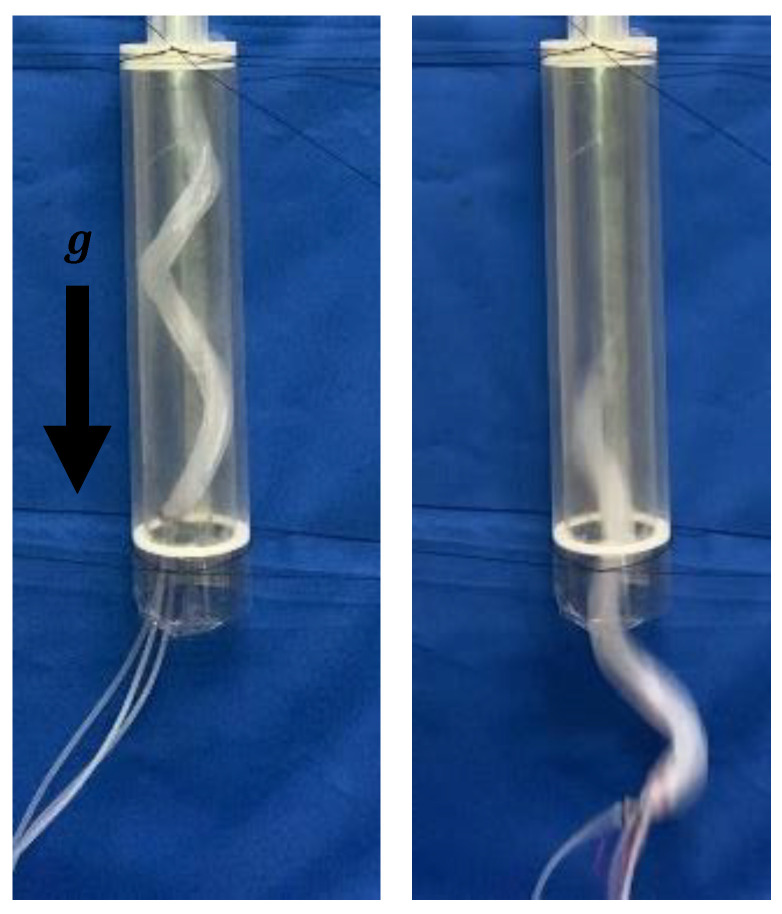
Experiment using a biased pitch device, driven by an air pressure of 0.14 MPa into a vertically fixed pipe with the same diameter as that of the 40A and the 20A.

**Figure 18 sensors-22-04997-f018:**
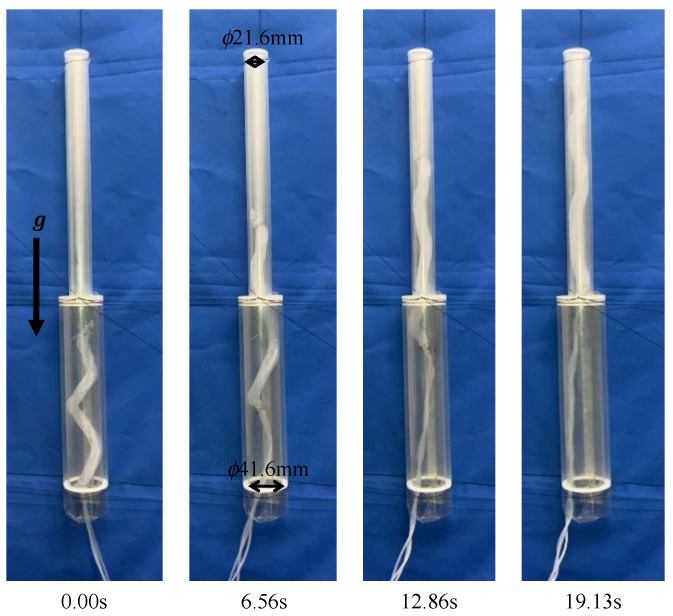
Experiment using a biased pitch device, driven by an air pressure of 0.15 MPa into a vertically fixed pipe with the same diameter as that of the 40A and the 20A.

**Figure 19 sensors-22-04997-f019:**
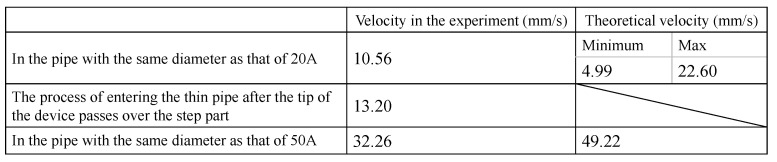
The velocities of the device.

**Figure 20 sensors-22-04997-f020:**
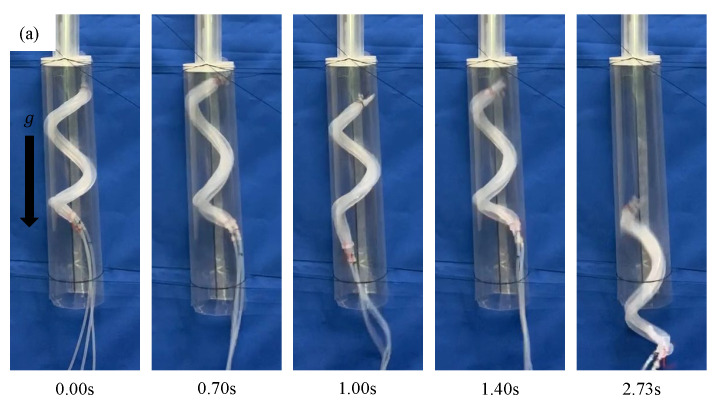
Experiment using a device with a uniform initial pitch of 320 mm driven into a vertically fixed pipe with the same diameter as that of the 50A and 20A pipes, where (**a**,**b**) show devices driven by air pressures of 0.17 MPa and 0.18 MPa, respectively.

**Figure 21 sensors-22-04997-f021:**
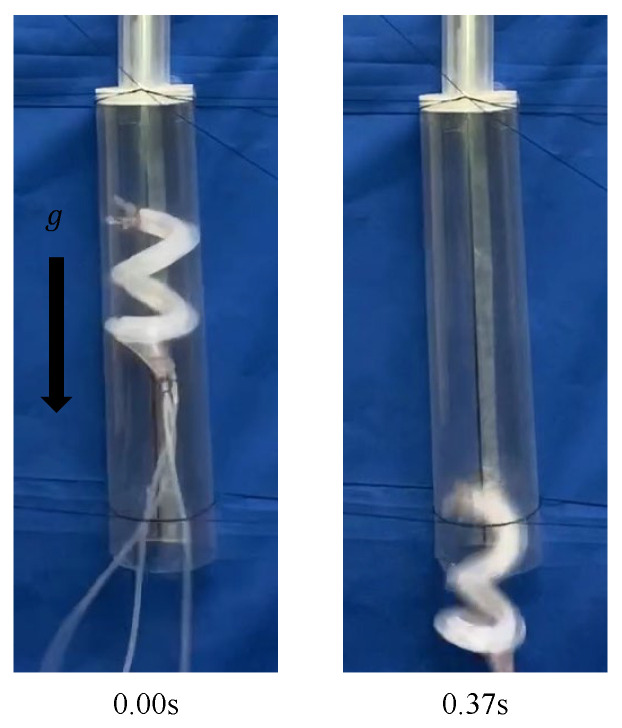
Experiment using a uniform pitch device driven by an air pressure of 0.20 MPa into a vertically fixed pipe with the same diameter as that of the 50A and 20A pipes.

## Data Availability

The datasets used and/or analyzed during the current study are available from the corresponding author on reasonable request.

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
