# Peer review of "Tapered, Twisted Bundled-Tube Locomotive Devices for Stepped Pipe Inspection"

_sensors, 2022, doi:10.3390/s22134997_

Round 1

Reviewer 1 Report

This paper presents the operation of the prototype of a soft robot designed to operate in pipelines. The robot is composed of a bundle of inflatable chambers, and the locomotion inducing pneumatic actuation consists of the differential inflation of the chambers that result into a spiral shape and net motion along the symmetry axis.

The paper focuses on a very specific operational scenario, in which the device has to negotiate a pipe diameter restriction, which could be the model of an obstruction or of a geometric change. 

The work is relatively narrow, as it focuses on the scenario just explained. However, I think that it has merit in offering some insights on possible operational constraints on this class of robotic devices. In my opinion, a work of this nature has merit even if results show strong limitations of the device, since this can be valuable for others working in the field and making decisions about future trends and designs. Indeed, it seems to be the case that for the specific scenario considered here, the operation is very fragile in the sense that a small variation of the input pressure can drastically change the desired outcome (passing through the restriction).   Specific comments follow, but my main concern is to expand the discussion about possible reasons for the device sensitivity to different parameters, with some insight on the viability of the design considering that a mature version of the device would have to operate autonomously in highly variable environments. Also, I am surprised about the limited number of references considering the vast literature on snake-like robots (the authors seem to refer just to a conference paper).

Points to be addressed:

1. The operation of the device seems to be very sensitive to some parameter, especially the input pressure. An insightful discussion about this point should be carried out, so that the reader could form an opinion about intrinsic limitations, applicability, and the possibility of overcoming limitations through future developments.

2. Refences should be expanded. The literature on snake like robots is vast, and additional references should be included.

3. Edits:

- Sections 3.2 and 3.3 have the same title

- The paragraph between lines 111-117 should be removed as it is part of the journal template.

- Line 12: "creatures" should be "living creatures".

- Line 13: the sentence "In particular, the locomotion of creatures with elongated trunks such as snakes is interesting." is quite generic. Interesting with respect to what? It should be better specified or perhaps removed as it doesn't seem to add any relevant information.

- Line 19, sentence "This makes it easy for them to become airtight, and thus highly resistant to the environment." should also be better qualified. In which sense are they better resistant to the environment? The sentence is also not clear in connection to the the one preceding it.

- Line 47: "[...] motors, The [...]" should be ""[...] motors, the [...]""

Reviewer 2 Report

This manuscript reports a twisted bundled tube locomotive device, which moves in a pipe using a helical bending motion. However, this manuscript must be significantly improved in its redaction and description of methods, materials, and discussion.

1.-English style and grammar must be improved.

2.-Abstract is confusing. This section should consider information of methods and materials. In addition, this section should add main results and conclusion. Also, abstract has terms 50A to 20A, whose parameter A was not described.

3.-Introduction section should include the main advantages and limitations of the soft robots reported in the literature. Furthermore, this section should consider the main advantages of the proposed soft robot in comparison with other reported in the literature.

4.-Description of method is confusing. The description of method must be significantly enhanced. All the parameters used in Equations (1)-(4) must be described.

5.-Authors should improve the description of the operation principle, dimensions, and materials of the soft robot. Figure 2 and 3 must be improved. The description of these figures must be enhanced.

6.-Experiments section has several mistakes in its redaction. For instance, this section has the sentences:

Materials and Methods should be described with sufficient details to allow others to replicate and build on published results. Please note that publication of your manuscript implicates that you must make all materials, data, computer code, and protocols associated with the publication available to readers. Please disclose at the submission stage any restrictions on the availability of materials or information. New methods and protocols should be described in detail while well-established methods can be briefly described and appropriately cited.

7.-The description of the experimental setup is poor. Authors must include a suitable description of the different stages of the experiments of the soft robot.

8.-Authors should include more experiments about performance of the soft robot.

9.-Discussion of the behavior of the soft robot must be improved.

10.-Which are the limitations of proposed soft robot?

11.-Conclusion is very poor. This section must be improved.

Reviewer 3 Report

The paper is devoted to a very interesting topic: the desgin of non-standard, non-classical locomotion systems. Most locomotion systems use wheels or legs, here, the authors present a new locomotor principle of a tube robot system which can be used, e.g., for in-pipe inspection or especially for missions in impassible region (e.g., after earthquakes).

The experiments in the paper are convincing, and the locomotor principle is really nice, because the system can move through pipes having sections with changing diameters.

But, unfortunately:

#1 The authors' work is theoretically based on equations and derivations presented in a previous published paper. Therefore, it is hard to follow the equations (1)-(4). They are from reference [19]. That's the point, that ...

#2 ... ALL references are cited in the introduction. This is a bit disappoiting, because there is no related work. Therefore, ...

#3 ... the reviewer suggests to have a closer inspection to the following references:

   [a] Incorporating tube-to-tube clearances in the kinematics of concentric tube robots
       Junhyoung Ha; Pierre E. Dupont
   [b] Torsional Kinematic Model for Concentric Tube Robots
       Pierre E. Dupont, Senior Member, IEEE, Jesse Lock, and Evan Butler

The investigations therein deal with the analysis of similar systems. So, please extend your references and check these papers and the references therein.

#4 The introduction starts a bit like "once upon a time ...". May you tighten the introduction.

#5 Are there any dynamical investigations to your system? What about friction? Are there any restrictions to the environmental contact of your robot?

#6 Special parameters \alpha=1.11, what's the reason of this choice? What about the given values of the diameters, will the robot perform his movement also in tubes of different diameters?

#7 Are there any hints for some control strategies? Controlling a given motion pattern, to achieve a desired locomotion gait in the tube?

#8 What about the efficiency ratio?

Summarizing: The paper is very nice, but needs major revisions!

Round 2

Reviewer 2 Report

This second version of manuscript has been improved considering the comments of reviewer. This revised manuscript can be accepted for publication in Sensors.

Reviewer 3 Report

The authors have spent a lot efforts to improve the presentation of the paper, due to the comments of the reviewers.